# Recreation and Alcohol Consumption in Sub-Saharan Africa: Addressing Gender and Age Differences in Urban Areas—Praia, Cabo Verde

**DOI:** 10.3390/ijerph191811175

**Published:** 2022-09-06

**Authors:** Daniela Alves, António Pedro Delgado, Miguel Amado, Isabel Craveiro, Zélia Santos, Alexander Goggins, Carolina Gasparinho, Artur Correia, Luzia Gonçalves

**Affiliations:** 1Unidade de Saúde Pública Global, Instituto de Higiene e Medicina Tropical, Universidade Nova de Lisboa (IHMT NOVA), Rua da Junqueira 100, 1349-008 Lisboa, Portugal; 2Global Health and Tropical Medicine, IHMT-NOVA, Rua da Junqueira 100, 1349-008 Lisboa, Portugal; 3Universidade de Cabo Verde, Zona K do Palmarejo Grande, Praia 7943-010, Capo Verde; 4Civil Engineering Research and Inovation for Sustainability, CERis, Instituto Superior Técnico, Universidade de Lisboa, Avenida Rovisco Pais, 1049-001 Lisboa, Portugal; 5Escola Superior de Tecnologia da Saúde de Lisboa (ESTeSL), Instituto Politécnico de Lisboa (IPL), Av. D. João II, 1990-096 Lisboa, Portugal; 6Universidade Intercontinental de Cabo Verde, n 1 Palmarejo, Praia, Capo Verde; 7Centro de Estatística e Aplicações da Universidade de Lisboa (CEAUL), Faculdade de Ciências da Universidade de Lisboa, 1749-016 Lisboa, Portugal

**Keywords:** recreation, leisure, alcohol consumption, urban planning, Cabo Verde

## Abstract

Background: Reducing alcohol consumption and improving urban planning in African cities are public health priorities. The aim of this study was to explore gender and age differences in recreational activity participation and its link with self-reported alcohol consumption in three urban areas of Praia. Methods: A questionnaire was applied to a probabilistic sample of 1912 adults, with a median age of 35.0 (IQR: 26.0–48.8) years, living in informal, transition, and formal areas of the capital of Cabo Verde. Results: More than 80% of the participants reported rarely or never participating in recreational activities. Going daily or weekly to the café was the most reported recreational activity, regardless of the urban area. Participation in recreational activities was higher in men than women, decreasing with age in both cases. Alcohol consumption was significantly higher in men than women (72.4% versus 47.4%, *p* < 0.001). Multiple logistic regression models showed that going at least once to the bar/nightclub (for men and women) and going to the café (for women) were associated with alcohol consumption. Furthermore, age (for women), in a protective way, and having children (for men) appeared to be associated with alcohol consumption. Conclusions: This study provides new data on the recreational environment in Praia and can contribute to the development of local and national public health policies and interventions in line with several SDGs to reduce alcohol consumption, enhance healthy leisure/recreation practices, and promote better living conditions for its inhabitants.

## 1. Introduction

“Good health and wellbeing” (SDG 3) and “gender equality” (SDG 5) are two of the 17 sustainable development goals (SDGs) included in the 2030 Agenda for Sustainable Development [1]. Alcohol consumption in Sub-Saharan Africa is a public health problem, which can be influenced by leisure and recreation activities, especially in more vulnerable populations [2,3]. In 2016, the estimated prevalence of alcohol dependence in Western Sub-Saharan Africa was 1168.1 (95% uncertainty interval: 1012.8–1335.2) cases per 100,000 people [4].

Numerous definitions of leisure and recreation are used in the literature [5,6,7,8], and it can sometimes be difficult to distinguish them because of overlapping meanings [9]. These concepts are often used interchangeably, although there are differences between them [10]. The definition of these concepts depends on the societal and cultural contexts, as well as individual perceptions [5,11].

Leisure has been described as the portion of an individual’s time which is not directly devoted or connected to work, responsibilities, or other obligatory forms of maintenance or self-care [11]. Thus, it may involve some form of participation in a voluntary activity, but it can be considered as a holistic state, spiritual experience, or absence of activity [9,11]. Recreation includes, among different definitions, voluntary activities (such as outdoor activities, sports, games, crafts, performing arts, fine arts, music, travel/tourism, hobbies, and social activities) engaged during an individual’s free time, having more connotations than leisure [9,10,11]. Leisure is a concept described in the literature as more related to an individual’s perspective, whereas recreation has a morally acceptable connotation for society as a whole [10,12].

The demand for leisure and recreational outdoor activities (physical or others) depends not only on individual determinants, such as sociodemographic characteristics and financial accessibility, but also on environmental factors, such as the availability and the accessibility to recreational infrastructures [13,14,15]. These determinants can restrict or allow the access to recreational activities, not only to the type of the activity, but also to the time that individuals spend on it [13,15].

Leisure and recreational activities can lead to increased levels of happiness and satisfaction, as well as promote a healthy lifestyle [5,7,16,17,18]. At the societal level, leisure and recreational activities can help strengthen family and community ties, facilitate integration, and preserve cultural identity [7,19]. They may also promote the socioeconomic development of a country and its inhabitants [4,7,20]. On the other hand, according to O’Sullivan [10], the choice of recreational activities may involve compulsion, discomfort, pain, or danger.

Some studies reported that recreation is associated with higher consumption of alcohol and other negative behaviors in younger people [17,21]. The health risks of drinking alcoholic beverages are known [22], and infrastructures for leisure and recreational activities usually sell these beverages (e.g., cafés, bars, and nightclubs). Many studies regarding leisure and recreation focused on the younger or older age groups, or on adults with specific diseases [5,6,17,23,24,25,26].

Worldwide, approximately 13.5% of total deaths in people between 20 and 39 years of age are alcohol-attributable [22]. According to the political declaration of the third high-level meeting of the General Assembly on the prevention and control of noncommunicable diseases, reducing the harmful use of alcohol is considered a public health priority in the action plan (2022–2030) [27]. However, to our knowledge, studies providing information on leisure and recreational activities of adult populations in African cities are lacking, as well as those exploring their relationship with alcohol consumption.

This study was implemented in Cabo Verde, a Sub-Saharan African archipelago of 10 islands (nine habited) with 27% of the urban population living in the capital, the city of Praia, Santiago Island [28], which is composed of 29 neighborhoods [29]. As many other African countries, Cabo Verde has witnessed a process of rapid urbanization [30], leading to a proliferation of informal neighborhoods characterized by spontaneous occupation with self-building of houses, without well-defined urban planning, lacking green spaces, public services, and basic infrastructures [31,32]. These informal urban areas are mostly inhabited by groups with greater social and economic vulnerabilities [31,32].

According to the World Health Organization (WHO) estimates, the average consumption per capita of pure alcohol in Cabo Verde reduced from 7.6 L in 2010 to 5.7 L in 2016 [22]. Despite the decrease, alcohol consumption is still one of the top 10 risk factors contributing to disability-adjusted life years (DALYs), occupying the seventh position in 2019 in this country [33]. According to the National Statistical Institute of Cabo Verde, the percentage of alcoholic beverages in household expenditure varied from 1.8% in 2002 to 4.0% in 2015 [34].

This paper aims to characterize some recreational activities (going to theater/cinema, restaurant, café, and bar/nightclub) by gender and age groups in three urban areas of the city of Praia, Cabo Verde, and to explore the link with the consumption of alcoholic beverages.

## 2. Materials and Methods

### 2.1. Study Design, Setting, and Participants

This work explores data collected from the research project, Urban Planning and Health Inequities—moving from macro to micro statistics (UPHI-STAT project) [32]. As shown in Figure 1, this study was developed in three urban areas of the city of Praia (Figure 1), between January and March 2014:(i)A formal area (Plateau neighborhood, the historic center), with well-established urban planning, public services, and infrastructures;(ii)A transition area (Palmarejo neighborhood), with characteristics of both formal and informal areas;(iii)An informal area (Vila Nova neighborhood), characterized by a lack of urban planning, public services, and basic infrastructures [32].

Figure 1 also shows some infrastructures for recreational activities (cultural spaces, restaurants, cafés, and bars/nightclubs) in Praia, in 2014. The city offers cultural official spaces, namely, the “Cine-Teatro” (Plateau neighborhood), the Auditório Nacional Jorge Barbosa (Varzea neighborhood), the Assembleia Nacional, and Centro Cultural Português (Achada de Santo António). Since December 2013, there is a cinema in the city’s shopping mall known as “Praia Shopping” (located in the Quebra Canela neighborhood and bordering the Palmarejo, Achada de Santo António and Prainha neighborhoods). The cafés and restaurants are scattered throughout the city, with a higher concentration in the Achada de Santo António and Plateau neighborhoods. The bars and nightclubs are also located throughout several neighborhoods of the city (two of them in the Plateau and the other two in the Palmarejo neighborhood) (Figure 1).

Sampling was performed by a random selection of geographical coordinates of the households, including adult participants (>18 years of age) living in the urban areas for at least 6 months [32]. On the basis of a confidence interval to a binomial proportion, the sample size was initially estimated as at least 1776 participants. To obtain a probabilistic sample, using geographic information system and statistical software, a sampling frame based on the geographic coordinates of households in each urban unit was built, and a proportional random sample was generated for each urban area as described previously [32]. The sample included 1912 adult participants—formal (*n* = 145), transition (*n* = 1144), and informal areas (*n* = 623).

### 2.2. Questionnaire

A questionnaire was applied by trained local interviewers in the three areas [32]. In this paper, five components were considered given a potential association with the consumption of alcoholic beverages, with some original variables and other recoded variables:(i)Space (urban area of residence in the city): formal, transition, and informal areas;(ii)Sociodemographic variables: gender (men, women), age (in years) recoded into age groups (15–25, 26–40, 41–65, and ≥66 years), marital status (single, married, civil union, divorced, and widowed), children (no or yes), academic qualifications (none or preschool, primary, secondary, and high school), and professional status (worker, unemployed, student, retired, and domestic);(iii)Self-reported consumption of alcoholic beverages (never or at least once) and type of alcoholic beverage consumed (beer, wine, liquors, “grog”, and whisky/gin/vodka);(iv)Frequency of participation in recreational activities (as “daily”, “weekly”, “monthly”, “rarely”, and “never”, recoded into a binary variable “never” versus “at least once”);(v)Type of recreation activities (theater/cinema, restaurant, café, and bar/nightclub).

### 2.3. Statistical Analysis

The statistical analysis was performed using the program SPSS, version 23 [35]. The descriptive analysis of qualitative variables is presented as absolute (*n*) and relative (%) frequencies. Asymmetric quantitative variables are described by the median and interquartile interval, denoted by IQR.

The associations between qualitative variables were determined using chi-square or Fisher exact tests. Using the self-reported alcoholic beverages consumption (0—never, 1—consumption of some type of alcoholic beverage) as the dependent variable, several binary logistic regression models stratified by gender were obtained. Adjusted models included variables from simple logistic regression models with *p* < 0.20 and other relevant variables to address confounding [36,37]. The variables included in these models were urban area, age (continuous variable), marital status, academic qualification, unemployment, children, and recreational activities: theater/cinema, restaurant, café, and bar/nightclub. Through these models, adjusted odds ratios (adj OR) and their 95% confidence intervals (95% CI) were obtained. The Hosmer–Lemeshow goodness-of-fit test and Nagelkerke *R*^2^ were obtained for several fitted models.

### 2.4. Ethical Considerations

The UPHI-STAT project was approved by the National Committee for Ethics in Research for Health (doc. N.52/2013), Cabo Verde, and Ethics Council of IHMT (doc n°.24-2013-PI), Portugal [26]. Written informed consent (two copies) was obtained from all participants.

## 3. Results

### 3.1. Sociodemographic Characteristics and Alcohol Consumption

Of the total participants, 1231 (64.4%) were women and 681 (35.6%) were men, and their median age was 35.00 years (IQR: 25.75–45.00) and 33.00 years (IQR: 27.00–50.00) (*p* = 0.001), respectively. More than half of the participants were living in the transition area (59.8%), while others were living in informal areas (32.6%), and a reduced percentage were in formal areas (7.6%).

Nearly 40% of overall participants reported that they were living in the city of Praia since their birth. The remaining participants (60.4%) presented a median length of residence of 16.0 years (IQR: 6.0–28.0). Considering the studied urban areas, 36.3% of participants had lived in the informal area since their birth, followed by 26.2% in the formal area, and a minor percentage of 8.0% in the transition area (*p* < 0.001). Analyzing previous generations, more than 68% of the participants reported that their parents or grandparents lived outside the city of Praia.

Table 1 presents the distribution of participants by area of residence, age group, marital status, children, academic qualification, job status, and self-reported consumption of alcoholic beverages.

As shown in Table 1, men and women are slightly equally distributed by areas (*p* = 0.411), although there was a significant difference (only at 5% level) between male and female in terms of age group (*p* = 0.041). A higher proportion of men and women was between 26 and 40 years of age (41.9% and 38.8%, respectively). Regarding marital status, more than 60% of men and women reported being single.

The percentage of men and women who reported having at least one child was 64.3% and 82.3% (*p* < 0.001), respectively. Differences in academic qualifications and professional status were observed by gender *(p* < 0.001), with a higher percentage of women (15.9%) having none/preschool qualifications, compared to men (6.2%), and with a lower percentage of women reporting being employed compared to men (40.7% versus 62.7%) (Table 1). For the younger age group (18–25 years), the percentage of those unemployed was 23.7%. Global consumption of alcoholic beverages was 56.3%, with a significant difference between men (72.4%) and women (47.4%), *p* < 0.001—Table 1.

### 3.2. Recreational Activities by Urban Areas

The frequency of participation in recreational activities (theater/cinema, restaurant, café, and bar/nightclub) by urban areas is presented in Table 2.

Globally, the majority of participants reported not participating in any of the described activities, such as going to the theater/cinema (63.9%), the restaurant (50.2%), the café (57.2%), and the bar/nightclub (59.6%). Participation in these activities varied significantly across urban areas (for all *p* < 0.001, Table 2).

Few participants reported going to the theater/cinema daily, weekly, or monthly. The monthly percentage was similar in the formal (12.7%) and the transition (14.3%) areas, but lower in the informal area (8.3%).

Going daily or weekly to the café was the most reported recreational activity by participants compared to the other activities, regardless of the urban area. Going to the café daily was reported by 10.6% of participants living in the formal area, a higher percentage compared with transition and informal areas (2.4% and 2.3%, respectively). Going to the café weekly was more frequent among participants of the transition area (10.7%).

Participation in bar/nightclub recreational activity (weekly and monthly) was higher in the transition area (15.2%), followed by informal (13.6%) and formal areas (9.1%). Monthly participation in this activity was more frequent in formal and transition areas, while, in the informal urban area, a higher percentage of participants reported going there weekly.

### 3.3. Recreational Activities by Gender

The frequency of participation (daily, weekly, monthly, rarely, or never) in recreational activities by gender is presented in Figure 2. Statistically significant differences were observed between men and women in the frequency of participation in the theater/cinema, restaurant, café, and bar/nightclub. Most of the individuals reported not participating in the studied recreational activities. Never going to the bar/nightclub was more reported by women (67.8%) compared to men (44.9%). Going rarely to the theater/cinema was reported by 34.2% and 26.5% of women and men, respectively. Men reported going more frequently to the restaurant and the bar/nightclub than women. Considering the recreational activity of going to a café, the same pattern was observed, except for the monthly participation which was more reported among women.

### 3.4. Recreational Activities by Age Group and Urban Areas

Figure 3 shows the percentage of participation in the recreational activities (daily, weekly, monthly, or rarely) among urban areas in each group.

As presented, significant differences were observed in the performance of all recreational activities among participants of the same age group living in different types of urban areas, with exception of those between 18–25 years of age, where no differences were detected for going out to the café, the restaurant, or the bar/nightclub.

Participants aged 26–40 years of the formal area had the highest rates of attendance to the café (78.8%), restaurant (77.1%), theater/cinema (73.5%), and bar/nightclub (71.4%). For all recreational activities, there was a decreasing gradient across the formal, transition, and informal areas in each age group. However, some exceptions occurred in terms of going to the bar/nightclub in two age groups (18–25 and 26–40 years). In these two groups, the attendance to the bar/nightclub was higher in the informal area (18–25 years: 57.6%; 26–40 years: 47.8%) compared to the transition area (44.9% and 44.8%, respectively). In the three urban areas, the percentage of going to the theater/cinema, restaurant, café, and bar/nightclub decreased with age.

### 3.5. Recreational Activities by Age Group and Gender

The frequency of participation (daily, weekly, monthly, or rarely) in the recreational activities between men and women in each age group is presented in Figure 4.

For all age groups under analysis, men reported going more frequently to the theater/cinema, restaurant, café, and bar/nightclub compared to women. The most frequent recreational activity reported by men and women was going to a restaurant (66.4% and 55.0%, respectively), both in the age range of 26–40 years old. Significant gender differences were observed within the four age groups, with two exceptions. The first exception was observed in the older participants (66+ years) regarding going to the theater/cinema and bar/nightclub. The second exception was found in the younger participants (18–25 years), with similar percentages of attendance to the theater/cinema, restaurant, and café. The largest discrepancies between men and women were observed in the age group of 41–65 years in going to the bar/nightclub, café, and restaurant, with percentage differences of 30.8%, 23.7% and 22.1%, respectively. Another gender discrepancy greater than 25.0% was observed in the older age group (66+) regarding going to a restaurant at least once.

### 3.6. Self-Reported Alcohol Consumption

As previously reported in Table 1, the alcoholic beverage consumption varied by gender (men: 72.4% and women: 47.4%; *p* < 0.001). Although this difference was also marked in subgroups, here we only add a global overview of the alcohol consumption by urban area, age group, and recreational activities (Appendix A).

Considering the alcohol consumption by urban area, no significant differences were found, with more than half of participants in each area reporting consuming some type of alcoholic beverage (formal: 57.2%, transition: 56.7%, informal: 55.4%; *p* = 0.842). Alcoholic beverage consumption also varied across age groups (*p* < 0.001).

More than half of participants between 18 and 65 years of age reported consuming some type of alcoholic beverage (aged 18–25 years: 59.0%, 26–40 years: 62.5% and 41–65: 50.9%), while those older than 66 years reported consuming less alcoholic beverage (38.9%).

Associations between self-reported alcohol consumption and recreational activities were also significant (*p* < 0.001 for all activities) (Appendix A). Percentages of self-reported alcohol consumption in participants who reported going at least once to each recreational activity were as follows: theater/cinema (70.7%), restaurant (69.9%), café (73.6%), and bar/nightclub (77.3%). About 51% of participants who reported never performing the studied recreational activities also reported never consuming alcoholic beverages (Appendix A). Regarding the type of alcoholic beverage (at least once), 83.9% reported consuming beer, along with 75.4% wine, 62.5% liquors, 39.6% “grog” (traditional Cabo Verdean drink—sugar cane brandy), and 38.1% whisky/gin/vodka. The percentages were statistically higher in men compared to women (except for wine and liquors), with the largest discrepancy observed in grog consumption (men: 70.4%; women: 13.6%) (Appendix B).

### 3.7. Factors Associated with Alcohol Consumption

Multiple logistic regression models, considering alcoholic beverage consumption as a binary variable, are presented in Table 3, stratified by gender.

Regarding recreational activities, controlling for sociodemographic variables, going out to the bar/nightclub at least once had significant associations with alcohol consumption for men and women. Furthermore, women who reported going to a café at least once were more likely to report alcoholic beverages consumption, compared to those who reported never going to a café (OR = 2.099, 95% CI: 1.456–3.026).

For men, it was also observed that having children increased the chance of alcohol consumption compared to men without children (OR = 2.995; 95% CI: 1.843–4.736). For women, an increase in age was associated with lower alcohol consumption (OR = 0.985; 95% CI: 0.976–0.995).

## 4. Discussion

This study contributes to a better understanding of how men and women of different age groups participate in recreational activities (theater/cinema, restaurant, café, and bar/nightclub), in three urban areas of Praia. Additionally, this study explores the link between these recreational activities and alcoholic beverage consumption. The absence of local studies about recreational activities in urban African settings, considering this urban gradient (informal, transition, and formal) imposes some difficulties to confront our findings with other studies. On the other hand, this study with adult participants, living in formal, transition, and formal areas, opens research lines at the local level to act on different neighborhoods of this heterogenous city, with a potential translation to other urban settings.

The focus of many African policies includes promoting health, education, and employment opportunities [38]. The World Leisure Organization argues that a focus on developing leisure and recreational activities should complement other social and health policies. Studies are crucial to better understand the leisure and recreational habits of these populations and, consequently, design policies [38].

In this study, the majority of participants (>2/3) reported that they rarely or never participated in the studied recreational activities. Our findings are in line with data obtained by the National Statistical Institute of Cabo Verde in 2015 on practices of culture, sport, and leisure of the Cabo Verdean population over 12 years of age [39]. According to this report, a great percentage of Cabo Verdeans reported never going to the theater (89.8%), the cinema (95.0%), or the restaurant/bar/nightclub (65.7%) [39]. For those living in Praia, the same report showed percentages of 82.7% and 88.8% of never going to the theater and cinema, respectively [39].

This city is both the political and the economic center of the country, attracting people from other parts of Santiago Island, other islands, and abroad [29,40,41]. Plateau (formal area) is the historical center of the city of Praia with more services and infrastructures, as well as fewer residents, compared to other neighborhoods [40]. The Cabo Verdean population is young. According to the National Statistical Institute of Cabo Verde [42], 45.2% of inhabitants are between 25 and 54 years old, with a national mean age of 28.01 years in 2015. Similarly, the average age in the city of Praia was 26.78 years old. Regarding marital status, our study showed that over 60.0% of the participants were single. The national statistics revealed the same pattern [43].

Our study found that most of the participants had secondary education. This finding corroborates the official data on the population living in Praia (where 41.0% have secondary education) [43]. One-quarter of our respondents had higher education, slightly higher than the official data of the city of Praia in 2014 (16.2%). Higher educational levels were found in men compared with women. Conversely, official country data in 2014 reported women as having higher education levels (10.0%) compared to men (7.7%) [43]. Cabo Verde is one of the Sub-Saharan African countries with the highest literacy rates [44], 91.0% for men and 82.0% for women [43].

There were more unemployed women (23.0%) than men (15.5%) in our study. However, the official unemployment statistics in Praia in 2014 reported percentages of 16.3% and 15.2% for men and women, respectively [43]. Younger age groups presented higher levels of unemployment (15–24 years: men—33.8%, women—38.5%; 25–44 years: men—14.2%, women—13.8%) [45]. The most recent data show that unemployment decreased between 2013 and 2017 [45], although it remained a problem for the country. According to Afrobarometer, Cabo Verde ranked fourth in a list of 32 African countries with the highest rates of unemployment [46]. According to Jalloh [38], elevated levels of unemployment may obscure the division of “work” and “leisure”.

In the formal area, younger participants (18–40 years) seemed to go to theater/cinema more frequently. Participants of the informal urban area reported a lower frequency of going to the theater/cinema. This result may be due to the more disadvantaged socioeconomic status of the inhabitants of the informal area and the availability of infrastructures for recreational activities. According to a study on recreation and leisure in multiethnic communities, minorities have limited access to theaters/cinemas due to socioeconomic factors, such as the lack of economic resources, overwork, and a lack of time [19]. Additionally, the same study suggested that the lack of information and restrictions on access to cultural activities is an explanation for the limited use [19]. In Iran, a study pointed out that the demand for going to the cinema depends on economic, cultural, social, individual, and environmental factors [14]. Of these factors, the environmental factors had a low impact, whereas the economic factors had a high impact on cinema attendance [14].

In the city of Praia, there has been an effort to attract more people to the cinema. Since 2014, various governmental and nongovernmental organizations have organized an international film festival in the city of Praia (“Plateau—Festival Internacional de Cinema da Praia”), which promotes national and regional movies from Portuguese-speaking African countries [47].

Our results showed that men reported going to the theater/cinema, restaurant, café, and bar/nightclub at a higher percentage than women, for all age groups. This may depend on family dynamics existing in the country, where important roles of the family are reserved to women [48]. It is also important to understand that the concept of the Cabo Verdean family is not limited to the nuclear family consisting of a mother, a father, and their children. In Cabo Verde, and African families in general, households can include grandparents, cousins, uncles, grandchildren, or even unrelated people. This central position of women in the family means greater responsibilities and concerns, which may explain lower levels of participation in recreational activities. It is also possible that women perform other recreational activities that have not been studied. In a study on the leisure activities of urban Africans, women reported that they had no leisure time, but they reported frequent visits to other family and friends [49]. While spending time with friends was the most referenced leisure activity, women still mentioned other activities such as going to the café or club, dancing, or participating in church activities [49]. Conversely, Massart [48] described Cabo Verdean men as having an active social life, which corroborates with the results of our study concerning restaurant, café, and nightclub attendance. A Cabo Verdean creole term *Konviviu* means leisure time spent by men among peers in public spaces or bars, as well as playing cards or watching football [48]. Data on culture, sport, and leisure in Cabo Verde also indicate a higher percentage of men’s participation in recreation and leisure activities compared to women’s participation [39]. Men are more involved in activities such as socializing with friends, watching television, playing/listening to music, and being with friends [39]. Greater participation of women compared to men in family outings and religious events was described [39]. Data from the National Statistical Institute of Cabo Verde indicated a low attendance percentage to the theater (women: 8.9%; men: 11.1%), cinema (women: 4.4%; men: 5.1%) and restaurant (women: 25.6; men: 43.0%) [39]. Yerkes et al. [50] showed that gender differences in leisure quality are lower in countries that promote gender equality by sharing household chores and where women have greater bargaining power. In line with this idea, at a local level, our findings may be used as a baseline to future studies, to address the impacts of the intensive work under the SDGs, particularly SDG 5 (gender equality), in Praia.

Participants belonging to the 26–40 age group, living in a formal urban area, reported going at least once to the restaurant, café, and/or bar/nightclub with higher frequency. A decreasing gradient across formal, transition, and informal urban areas was observed for recreational/leisure activities studied. Pressman et al. [51] analyzed sociodemographic predictors of scores for a leisure activity that included the frequency of meals out with friends/family for adults in Pittsburgh (19–89 years), overserving that income and education were significant predictors of participation in activities as included in a score, regardless of age, gender, or ethnicity [51]. Other studies conducted in European and United States contexts suggest that older people living in disadvantaged neighborhoods with high levels of crime and low socioeconomic status face greater barriers and are less involved in social activities [52,53,54].

Another factor that may partly explain the low reported rates of recreational activities is the climate of insecurity in the city. In 2015, the city of Praia was the municipality of Cabo Verde with the highest crime rate in the country, with 10.369 occurrences registered by Cabo Verdean Police [42]. Under the UPHI-STAT project, most participants reported that robbery and violence in the neighborhood where they live was a serious problem and that there was a need for greater security to improve their quality of life [32,55]. Afrobarometer [46] indicated that unemployment, insecurity, and poverty are the main problems in Cabo Verde. Thus, the perception of insecurity throughout the city may partly explain the low rates of participation in recreational activities, such as going to the theater/cinema, restaurant, café, and bar/nightclub, particularly for women.

Self-reported alcohol consumption did not differ significantly across the three urban areas. However, there was a greater vulnerability to alcohol consumption and associated problems in residents of informal urban areas [56,57]. On the other hand, significant differences were observed according to gender (higher in men compared to women: 72.4% and 47.4%, respectively) and age group (higher consumption in the youngest). Similarly, a recent study aiming to examine the prevalence and sociodemographic correlation of substance use in four Sub-Saharan African countries (South Africa, Kenya, Ghana, and Burkina Faso) reported a higher consumption in men compared to women (60.3% versus 29.3%) [58].

In particular, the fitted models showed that participants who reported going at least once to the café (for women) or bar/nightclub (for men and women) were more likely to be associated with alcohol consumption. Moreover, alcohol consumption was also associated with having children (for men) and age for women.

According to the reported perceptions of focus group participants, the alcohol consumption in Cabo Verde is pervasive, with men and women of all social classes and ages consuming alcohol regularly [55]. Data from WHO showed a decrease in alcohol consumption in Cabo Verde between 2010 and 2016 [22]. Currently, this consumption is below the average of the African region (6.3 L pure alcohol per capita). In this study, the type of beverage most consumed was beer, followed by wine, which corroborates the WHO report [59]. Despite positive developments in recent years, alcohol consumption in Cabo Verde remains high, being one of the top 10 risk factors contributing to disability-adjusted life years (DALYs), moving from sixth position in 2009 to seventh position in 2019 [33].

In Africa, among other factors, aggressive marketing activities by the alcohol industry and low regulation of alcohol companies (leading to greater availability and distribution) negatively influence national alcohol policies, and the degree of social acceptance of drinking has been described in the literature [2,3,60,61]. The relationship between socioeconomic characteristics and alcohol consumption has been previously reported [22,57,60,62]. Physical environments (e.g., private spaces, public spaces, or the number of drinking locations) may encourage or discourage heavy drinking, but diverse individual and social environments are also involved in high alcohol consumption in men and women [63].

Some limitations of this cross-sectional study are present in this analysis. In a cross-sectional study, it is impossible to separate a presumed cause and its possible effect; thus, to analyze and better understand the dynamic between recreational activities and alcohol consumption, studies over a period of time may bring solid findings [64]. Secondly, self-reported data were obtained by a questionnaire, which is subject to well-known biases [65]. Memory bias and social-desirability bias are expected, considering the potential stigma associated with alcohol consumption [66,67]. Despite these limitations, our study contribute new data in an African urban setting with very few previous studies.

## 5. Conclusions

In the studied urban areas of the capital of Cabo Verde, a high percentage of participants reported that they rarely or never participated in recreational activities such as going to the theater/cinema, restaurant, coffee shops, and/or bar/nightclub. The attendance to the theater/cinema was low, while going to the café or restaurant was the most frequently reported recreational activity across all age groups, urban areas, and genders. A decreasing trend between formal, transition, and informal urban areas was found in all age groups, for the studied recreational activity. Men participated more frequently in recreation activities than women, and participation decreased with age, for both. Furthermore, going at least once to the bar/nightclub seemed to explain the self-reported alcohol consumption. Additionally, other significant associations were found for men, such as having children. For women, age and going to the café were also associated with alcohol consumption. These results shed a light on the recreational environment in Praia. Future studies should be conducted, considering also other recreational activities and their relationship with alcohol consumption, as well as their impacts on health and wellbeing, in different urban areas of other cities in Cape Verde and other African urban settings.

Moreover, at a local level, this study can contribute to the development of recommendations, design of public health interventions, and formulation of public policies, in line with several SGDs, to reduce alcohol consumption, enhance healthy leisure/recreation practices, and promote better living conditions for its inhabitants.

## Figures and Tables

**Figure 1 ijerph-19-11175-f001:**
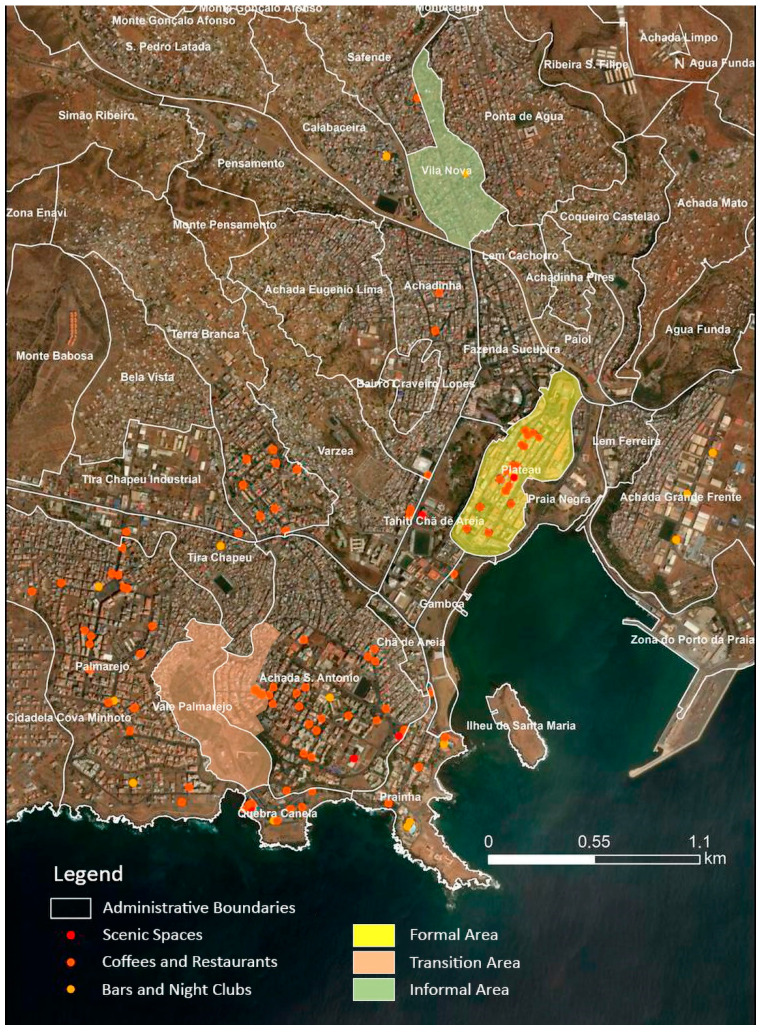
Recreational infrastructures in the city of Praia in 2014.

**Figure 2 ijerph-19-11175-f002:**
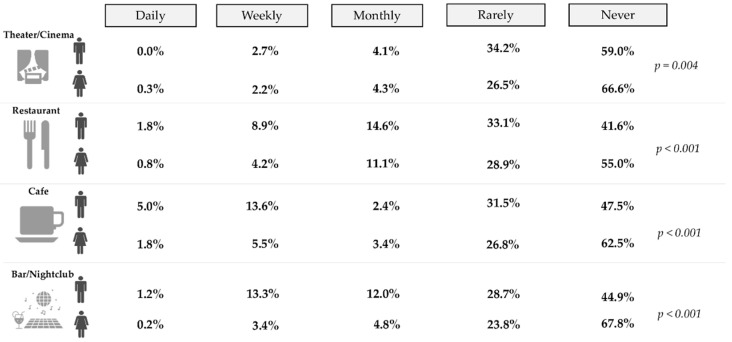
Frequency of the recreational activities by gender (*p*, *p*-value).

**Figure 3 ijerph-19-11175-f003:**
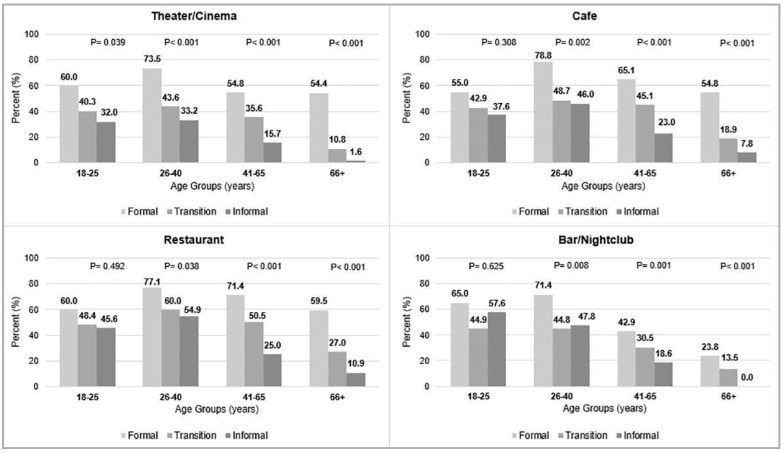
Frequency of the recreational activities among urban area in each age group (*p, p*-value).

**Figure 4 ijerph-19-11175-f004:**
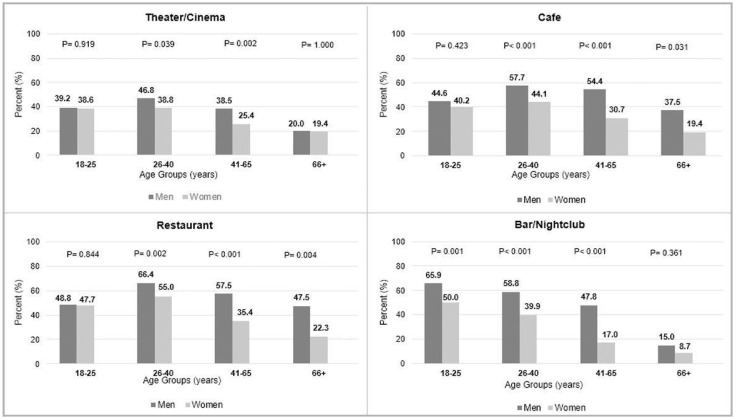
Frequency of the recreational activities between men and women in each age group (*p*, *p*-value).

**Table 1 ijerph-19-11175-t001:** Urban area of residence, sociodemographic characteristics, and alcohol consumption of participants by gender.

Variable	Total (*n* = 1912)	Men (*n* = 681)	Women (*n* = 1231)	*p*
*Urban Area of residence, n (%)*				0.411
Formal	145 (7.6)	56 (8.2)	89 (7.2)	
Transition	1144 (59.8)	415 (60.9)	729 (59.2)	
Informal	623 (32.6)	210 (30.8)	413 (33.5)	
*Age Groups (years), n (%)*				0.041
18–25	434 (23.0)	167 (24.9)	267 (22.0)	
26–40	752 (39.9)	281 (41.9)	471 (38.8)	
41–65	554 (29.4)	182 (27.2)	372 (30.6)	
≥66	144 (7.6)	40 (6.0)	104 (8.6)	
*Marital Status, n (%)*				0.006
Single	1189 (62.5)	436 (64.6)	753 (61.4)	
Married	364 (19.1)	132 (19.6)	232 (18.9)	
Civil union	227 (11.9)	78 (11.6)	149 (12.2)	
Divorced	44 (2.3)	17 (2.5)	27 (2.2)	
Widowed	77 (4.1)	12 (1.8)	65 (5.3)	
*Children, n (%)*				<0.001
No	460 (24.1)	243 (35.7)	217 (17.7)	
Yes	1450 (75.9)	438 (64.3)	1012 (82.3)	
*Academic qualifications, n (%)*				<0.001
None or preschool	236 (12.4)	42 (6.2)	194 (15.9)	
Primary	470 (24.8)	152 (22.4)	318 (26.1)	
Secondary	722 (38.1)	294 (43.2)	428 (35.2)	
High school	469 (24.7)	192 (28.2)	277 (22.8)	
*Professional status, n (%)*				<0.001
Worker	926 (48.5)	426 (62.7)	500 (40.7)	
Unemployed	388 (20.3)	105 (15.5)	283 (23.0)	
Student	283 (14.8)	100 (14.7)	183 (14.9)	
Retired	148 (7.8)	48 (7.1)	100 (8.1)	
Domestic	164 (8.6)	0 (0.0)	164 (13.3)	
*Consumption of alcoholic beverages, n (%)*				<0.001
No	835 (43.7)	188 (27.6)	647 (52.6)	
Yes	1077 (56.3)	493 (72.4)	584 (47.4)	

*p*, *p*-value.

**Table 2 ijerph-19-11175-t002:** Frequency of leisure/recreational activities by urban areas.

Variable	Total (*n* = 1912)	Formal (*n* = 145)	Transition (*n* = 1144)	Informal (*n* = 623)	*p*
*Theater/Cinema n (%)*					<0.001 ^f^
Daily	4 (0.2)	1 (0.7)	3 (0.3)	0 (0.0)	
Weekly	45 (2.4)	3 (2.1)	34 (3.0)	8 (1.3)	
Monthly	81 (4.3)	5 (3.5)	69 (6.1)	7 (1.1)	
Rarely	556 (29.2)	76 (53.9)	347 (30.5)	133 (21.4)	
Never	1216 (63.9)	56 (39.7)	686 (60.2)	474 (76.2)	
*Restaurant n (%)*					<0.001
Daily	22 (1.2)	6 (4.1)	11 (1.0)	5 (0.8)	
Weekly	111 (5.8)	4 (2.8)	87 (7.6)	20 (3.2)	
Monthly	235 (12.3)	18 (12.7)	163 (14.3)	54 (8.7)	
Rarely	579 (30.4)	69 (48.6)	350 (30.7)	160 (25.7)	
Never	956 (50.2)	45 (31.7)	528 (46.4)	383 (61.6)	
*Café n (%)*					<0.001
Daily	56 (2.9)	15 (10.6)	27 (2.4)	14 (2.3)	
Weekly	160 (8.4)	11 (7.8)	122 (10.7)	27 (4.3)	
Monthly	58 (3.0)	6 (4.3)	34 (3.0)	18 (2.9)	
Rarely	541 (28.4)	59 (41.8)	336 (29.5)	146 (23.5)	
Never	1087 (57.2)	50 (35.5)	620 (54.4)	417 (67.0)	
*Bar/Nightclub n (%)*					<0.001 ^f^
Daily	10 (0.5)	0 (0.0)	3 (0.3)	7 (1.1)	
Weekly	132 (6.9)	5 (3.5)	79 (6.9)	48 (7.7)	
Monthly	140 (7.4)	8 (5.6)	95 (8.3)	37 (5.9)	
Rarely	486 (25.5)	54 (38.0)	305 (26.8)	127 (20.4)	
Never	1135 (59.6)	75 (52.8)	657 (57.7)	403 (64.8)	

*p*, *p*-value; ^f^, Fisher exact test.

**Table 3 ijerph-19-11175-t003:** Multiple logistic regression models for consumption of alcoholic beverages by gender.

Variables	Men			Women		
	*p*	adj OR	95% CI	*p*	adj OR	95% CI
*Urban Area of Residence*						
Formal ^(a)^	0.822			0.146		
Transition	0.770	1.115	0.538–2.307	0.340	1.297	0.760–2.214
Informal	0.587	1.238	0.574–2.670	0.093	1.613	0.923–2.818
Age (years)	0.627	0.996	0.981–1.012	0.004	0.985	0.976–0.995
*Marital Status*						
Without partner ^(a)^						
With partner	0.085	0.673	0.429–1.056	0.494	1.101	0.835–1.451
*Children*						
No ^(a)^						
Yes	<0.001	2.955	1.843–4.736	0.076	1.394	0.966–2.010
*Academic Qualification*						
Lower ^(a)^						
Higher	0.191	1.344	0.863–2.094	0.843	0.969	0.712–1.320
*Unemployed*						
No ^(a)^						
Yes	0.573	1.163	0.688–1.965	0.661	0.935	0.692–1.263
*Theater/Cinema*						
Never ^(a)^						
At least once	0.334	1.282	0.774–2.123	0.969	1.007	0.709–1.431
*Restaurant*						
Never ^(a)^						
At least once	0.577	0.852	0.487–1.493	0.111	1.326	0.937–1.877
*Café*						
Never ^(a)^						
At least once	0.231	1.390	0.811–2.383	<0.001	2.099	1.456–3.026
*Bar/Nightclub*						
Never ^(a)^						
At least once	<0.001	3.352	2.056–5.465	<0.001	2.327	1.646–3.290

^(a)^, Reference category; *p*, *p*-value; adj OR, adjusted odds ratio; CI, confidence interval. Hosmer–Lemeshow test: men = 0.294; women = 0.921; Nagelkerke *R*^2^: men = 0.179; women = 0.186.

## Data Availability

The data presented in this study are available on request from the corresponding author. The data are not publicly available due to ethical requirements.

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
