# Peer review of "Recreation and Alcohol Consumption in Sub-Saharan Africa: Addressing Gender and Age Differences in Urban Areas—Praia, Cabo Verde"

_ijerph, 2022, doi:10.3390/ijerph191811175_

Round 1
Reviewer 1 Report
Dear Authors,
Thank you for your manuscript. The topic is important and relevant to public health. The paper is interesting and well-written. I have only a few minor comments.
Please provide the mean or median age of the study participants in the abstract.
The description of the study questionnaire is incomplete. In the Results section, variables appear first with the response options not mentioned in the Methods section (academic qualification, professional status, etc.). All questions should be presented in the Methods section with the response options (frequency of recreation activities, etc.). Also, I would recommend changing the "Health outcome indicator measure" (line 140) into "Self-reported alcohol consumption", as no health outcomes were investigated in this study.
Please provide study limitations at the end of the Discussion section.
Author Response
Dear Reviewer 1. Thank you for your interest in the manuscript and for your important considerations.
1) Please provide the mean or median age of the study participants in the abstract.
Response: The median age of the participants was included in the abstract.
Abstract - Lines 22-24: “Methods: A questionnaire was applied to a probabilistic sample of 1912 adults, with a median age of 35.0 (IQR: 26.0 - 48.8) years, living in informal, transition, and formal areas of the capital of Cabo Verde.”
2) The description of the study questionnaire is incomplete. In the Results section, variables appear first with the response options not mentioned in the Methods section (academic qualification, professional status, etc.). All questions should be presented in the Methods section with the response options (frequency of recreation activities, etc.). Also, I would recommend changing the "Health outcome indicator measure" (line 140) into "Self-reported alcohol consumption", as no health outcomes were investigated in this study.
Response: Thank you for this important contribution. The more relevant variables for this work were described in the Methods section. Considering your suggestion, the "Health outcome indicator measure" was replaced by the “self-reported consumption of alcoholic beverages”.
Methods – Lines 169-188: “A questionnaire was applied by trained local interviewers in the three areas [31]. In this paper, five components were considered given a potential association with the consumption of alcoholic beverages, with some original variables and other recoded variables, namely:
i) Space (urban area of residence in the city): formal, transition, and informal areas;
ii) Sociodemographic variables: gender (men, women), age (in years) and recoded into age groups (15-25, 26-40, 41-65 and ≥66 years), marital status (single, married, civil union, divorced and widowed), children (no or yes), academic qualifications (none and preschool, primary, secondary, high school) and professional status (worker, unemployed, student, retired and domestic);
iii) Self-reported consumption of alcoholic beverages (never or at least once), and type of alcoholic beverage consumed (beer, wine, liquors, “grog” and whisky/gin/vodka);
iv) Frequency of participation in recreational activities (as “daily”, “weekly”, “monthly”, “rarely”, and “never”; or recoded into a binary variable “never” versus “at least once”);
v) Type of recreation activities (theater/cinema, restaurant, cafe, and bar/nightclub).”
3) Please provide study limitations at the end of the Discussion section
Response: Limitations of the study were included in the discussion.
Discussion - Lines 559-576: “Some limitations of this cross-sectional study are present in this analysis. In a cross-sectional study, it is impossible to separate a presumed cause and its possible effect, thus, to analyze and better understand the dynamic between recreational activities and alcohol consumption, studies over a period of time may bring solid findings [64]. Second, self-reported data were obtained by a questionnaire, which is subject to well-known biases [65]. Memory bias and social-desirability bias are expected, considering the potential stigma associated to alcohol consumption [66,67]. Despite these limitations, our study added new data in an African urban setting with very few studies.”
Best Regards
Daniela Alves and co-authors

Reviewer 2 Report
Lines 19-21: Split this paragraph in two, so as to distinguish the background and the aim of the study: (1) Background.......; (2) Aim of the study:.....
Line 38: The quote "leave no one behind" in this sentence (The 2030 Agenda for Sustainable Development has a principle "leave no one behind") is not meaningful here
Lines 40-41: For a health issue to be considered a public health problem, there is a level of prevalence. What it is with regard to alcohol consumption?
Line 60: I would rather say financial accessibility
Lines 108-113: It will be quite interesting if the 3 areas are clearly shown on the figure 1
Lines 129-131: Since you've adopted a quantitative approach, it is critical to provide details on participants selection process: Justify the choice of the target population, i.e. the adults; How have you come to determine the sampling size of 1912 adults? How many were selected in each of the 3 urban areas (formal / transition / informal areas)?
Line 138: one important variable is missing among your sociodemographic variables: income or salary. If this is difficult to get from inhabitants, you can use a composite or aggregate variable such as standard of living.
Line 189: You wrote: '' men and women are equally distributed by areas''. I suggest: ''men and women are slightly equally distributed by areas''
For an original article like this, it would have been nice if you could draw 2 figures: one showing profile of women involved in leisure and Recreational activities; and a profile of men also involved in Recreational activities
Author Response
Reviewer 2
Dear Reviewer 2. Thank you for your interest in the manuscript and for your important considerations.
1) Lines 19-21: Split this paragraph in two, so as to distinguish the background and the aim of the study: (1) Background.......; (2) Aim of the study:.....
Response: The paragraph was changed, as suggested.
Lines 19-23. “Abstract: (1) Background: Reducing alcohol consumption and improving urban planning in African cities are public health priorities. (2) Aims: to explore gender and age differences in recreational activities participation and their links with self-reported alcohol consumption in three urban areas of Praia”
2) Line 38: The quote "leave no one behind" in this sentence (The 2030 Agenda for Sustainable Development has a principle "leave no one behind") is not meaningful here
Response: Thanks for the suggestion, the sentence in the introduction has been modified. Introduction - Lines 39-41: " “Good health and well-being" (SDG 3) and "gender equality" (SDG5) are two of the 17 sustainable development goals (SDG) included in the 2030 Agenda for Sustainable Development [1].”
3) Lines 40-41: For a health issue to be considered a public health problem, there is a level of prevalence. What it is with regard to alcohol consumption?
Response: Thanks for the comment. Information on estimated prevalence in the Western Sub-Saharan Africa region was included. Introduction - Lines 43-45: “In 2016, the estimated prevalence of alcohol dependence in Western sub-Saharan Africa was 1168.1 (95% uncertainty interval: 1012.8 – 1335.2) cases per 100,000 people [4].”
4) Line 60: I would rather say financial accessibility
Response: Thank you for your suggestion. It has been included in the manuscript. Introduction - Lines 78-81: “The demand for leisure and recreational outdoor activities (physical or others) depends not only on individual determinants, such as sociodemographic characteristics, financial accessibility but also on environmental factors, namely the availability and the accessibility to recreational infrastructures [13–15].”
5) Lines 108-113: It will be quite interesting if the 3 areas are clearly shown on the figure 1
Response. Thank you for your suggestion. Figure 1 was changed, and the three urban areas studied were identified by colors: Formal Area – Plateau neighborhood; Transition Area – Palmarejo neighborhood and Informal Area – Vila Nova neighborhood. Figure 1. Recreational infrastructures in the city of Praia in 2014
6) Lines 129-131: Since you've adopted a quantitative approach, it is critical to provide details on participants selection process: Justify the choice of the target population, i.e. the adults; How have you come to determine the sampling size of 1912 adults? How many were selected in each of the 3 urban areas (formal / transition / informal areas).
Response: Thank you for your comment. More information about sampling was included in the methods.
Methods: - Lines 155-167: “Sampling was performed by a random selection of geographical coordinates of the households, including adult participants (>18 years of age) living in the urban areas for at least six months [32]. Based on a confidence interval to a binomial proportion, the sample size was initially estimated in at least 1776 participants. To obtain a probabilistic sample, using Geographic Information System and statistical software, a sampling frame based on the geographic coordinates of households in each urban unit was built and a proportional random sample was generated for each urban area as described previously [32]. The sample included 1912 adult participants – formal (n = 145), transition (n = 1144), informal areas (n = 623).”
7) Line 138: one important variable is missing among your sociodemographic variables: income or salary. If this is difficult to get from inhabitants, you can use a composite or aggregate variable such as standard of living.
Response: Thank you for your comment. We understand the importance of the income or salary variable in this study. Unfortunately, income, salary or standard of living variable were not included in the initial design of the project.
8) Line 189: You wrote: '' men and women are equally distributed by areas''. I suggest: ''men and women are slightly equally distributed by areas''
Response: Thank you for the suggestion, which was included in the manuscript. Results - Lines 247-249: “As shown in Table 1, men and women are slightly equally distributed by areas (p = 0.411)”.
9) For an original article like this, it would have been nice if you could draw 2 figures: one showing profile of women involved in leisure and Recreational activities; and a profile of men also involved in Recreational activities. Response: Thank you for your suggestion. In the Results, a figure (Figure 2) was included with a description of the data on the frequency of recreational activities by gender Results – Lines 286-295
“3.3. Recreational activities by gender
The frequency of participation (daily, weekly, monthly, rarely, or never) in recreational activities by gender is presented in Figure 2. Statistically significant differences were observed between men and women in the frequency of participation in the theater/cinema, restaurant, cafe, and bar/nightclub. Most of the individuals reported not participating in the studied recreational activities. Never going to the bar/nightclub was more reported by women (67.8%) compared to men (44.9%). Going rarely to the theater/cinema was reported by 34.2% and 26.5% of women and men, respectively. Men reported to go more frequently to the restaurant and the bar/nightclub than women. Considering the recreational activity of going to a cafe, the same pattern was observed, excepting the monthly participation which was more reported among women.
Figure 3. Frequency of the recreational activities by gender (p, p-value).”
Best Regards
Daniela Alves and co-authors

Reviewer 3 Report
The research topic is judged to be creative.
Research in Africa is very interesting because it is difficult to select subjects and collect data.
This study is an interesting topic in the field of recreation, and it is considered to be of great help academically, especially since it was studied in Africa.
However, it is a well-known fact that leisure and recreation provide many benefits to people. Although new research results were expected, it is judged that the researcher's argument is insufficient in the discussion.
Nevertheless, this study is judged to be of great value as a study on leisure and recreation for Africans.
The content of the discussion is judged to be general. Differences between this study and previous studies or a creative researcher's argument are required.
In conclusion, I hope to include suggestions for follow-up studies.
Author Response
Reviewer 3
Dear Reviewer 3. Thank you for your interest in the manuscript and for your important considerations. 1) The research topic is judged to be creative. Research in Africa is very interesting because it is difficult to select subjects and collect data. This study is an interesting topic in the field of recreation, and it is considered to be of great help academically, especially since it was studied in Africa. However, it is a well-known fact that leisure and recreation provide many benefits to people. Although new research results were expected, it is judged that the researcher's argument is insufficient in the discussion. Nevertheless, this study is judged to be of great value as a study on leisure and recreation for Africans. The content of the discussion is judged to be general. Differences between this study and previous studies or a creative researcher's argument are required. In conclusion, I hope to include suggestions for follow-up studies. Response: Thanks for the commentary. To our knowledge there are no studies in an African context that would allow comparison of results, considering the gradient: informal, transition and formal. This knowledge of the dynamics of recreational practices and the link with alcohol consumption may contribute to the development of leisure policies that may complement other social and health policies in this city. To complement the discussion the following text has been added and follow-up studies were included in the conclusions:
Discussion – Lines 569-576: “The absence of local studies about recreational activities in urban African settings, considering this urban gradient (informal, transition, and formal) imposes some difficulties to confront our findings with other studies. On the other hand, this study with adults’ participants, living in formal, transition and formal areas, opens research lines at the local level to act on different neighborhoods of this heterogenous city, with a potential translation to other African urban settings.” Conclusion - Lines 590-593: “Future studies should be conducted, considering also other recreational activities and their relationship with alcohol consumption and their impacts on health and well-being in different urban areas of other cities in Cape Verde and other African settings. “
Best Regards
Daniela Alves and co-authors
